# Habitual Miso (Fermented Soybean Paste) Consumption Is Associated with a Low Prevalence of Sarcopenia in Patients with Type 2 Diabetes: A Cross-Sectional Study

**DOI:** 10.3390/nu13010072

**Published:** 2020-12-28

**Authors:** Fuyuko Takahashi, Yoshitaka Hashimoto, Ayumi Kaji, Ryosuke Sakai, Yuka Kawate, Takuro Okamura, Noriyuki Kitagawa, Hiroshi Okada, Naoko Nakanishi, Saori Majima, Takafumi Senmaru, Emi Ushigome, Masahide Hamaguchi, Mai Asano, Masahiro Yamazaki, Michiaki Fukui

**Affiliations:** 1Department of Endocrinology and Metabolism, Graduate School of Medical Science, Kyoto Prefectural University of Medicine, 465 Kajii-cho, Kawaramachi-Hirokoji, Kamigyo-ku, Kyoto 602-8566, Japan; fuyuko-t@koto.kpu-m.ac.jp (F.T.); kaji-a@koto.kpu-m.ac.jp (A.K.); sakaryo@koto.kpu-m.ac.jp (R.S.); yukawate@koto.kpu-m.ac.jp (Y.K.); d04sm012@koto.kpu-m.ac.jp (T.O.); nori-kgw@koto.kpu-m.ac.jp (N.K.); conti@koto.kpu-m.ac.jp (H.O.); naoko-n@koto.kpu-m.ac.jp (N.N.); saori-m@koto.kpu-m.ac.jp (S.M.); semmarut@koto.kpu-m.ac.jp (T.S.); emis@koto.kpu-m.ac.jp (E.U.); mhama@koto.kpu-m.ac.jp (M.H.); maias@koto.kpu-m.ac.jp (M.A.); masahiro@koto.kpu-m.ac.jp (M.Y.); michiaki@koto.kpu-m.ac.jp (M.F.); 2Department of Diabetology, Kameoka Municipal Hospital, 1-1 Noda, Shinochoshino, Kameoka-City, Kyoto 621-8585, Japan; 3Department of Diabetes and Endocrinology, Matsushita Memorial Hospital, 5-55 Sotojima-cho, Moriguchi 570-8540, Japan

**Keywords:** fermented soy foods, miso, sarcopenia, type 2 diabetes

## Abstract

Insulin resistance is a risk of sarcopenia, and the presence of sarcopenia is high in patients with type 2 diabetes (T2DM). It has been reported that habitual miso soup consumption was associated with lower insulin resistance. However, the association between habitual miso consumption and the presence of sarcopenia in patients with T2DM, especially sex difference, was unclear. In this cross-sectional study, 192 men and 159 women with T2DM were included. Habitual miso consumption was defined as consuming miso soup regularly. Having both low skeletal muscle mass index (<28.64% for men, <24.12% for women) and low adjusted hand grip strength (<51.26% for men, <35.38% for women) was defined as sarcopenia. The proportions of sarcopenia were 8.7% in men and 22.6% in women. The proportions of habitual miso consumption were 88.0% in men and 83.6% in women. Among women, the presence of sarcopenia was lower in the group with habitual miso consumption (18.8% versus 42.3%, *p* = 0.018); however, there was no association between habitual miso consumption and the presence of sarcopenia in men. Habitual miso consumption was negatively associated with the presence of sarcopenia in women (adjusted odds ratio (OR), 0.20 (95% confidence interval (CI): 0.06–0.62), *p* = 0.005) but not in men. This study indicated that habitual miso consumption was associated with the presence of sarcopenia in women but not in men.

## 1. Introduction

The number of older patients with type 2 diabetes (T2DM) in Japan is increasing, since the aging of the adult population is increasing [1]. These patients often complicate with sarcopenia, which is characterized as loss of muscle mass, power, and function [2]. In the last decade, sarcopenia has been reported as a risk factor for fatty liver, cardiovascular diseases, and death [3,4,5]. In insulin-resistant conditions including T2DM, insulin-stimulated glucose disposal is severely impaired in the skeletal muscle [6]. Insulin resistance and oxidative stress are reported to be a cause of sarcopenia [7,8,9], and they are related to vascular modification [10], chronic inflammation [11], and lipid inflation in muscles [12,13]. Moreover, low skeletal muscle mass, defined as the weight-adjusted absolute muscle mass, has been described as a risk factor for T2DM [14] and to be associated with insulin sensitivity [15]. Furthermore, low handgrip strength, defined as weight-adjusted handgrip strength, has been described as a risk factor for T2DM [16]. Therefore, it is useful to determine the percentage of muscle mass or handgrip strength per body weight in patients with diabetes.

Miso soup is a traditional Japanese dish, made from a fermented soybean food called miso, and is consumed widely in Japan. It includes vitamins, minerals, vegetable proteins, microorganisms, salts, carbohydrates, and fat [17]. A previous study has revealed that miso can suppress the development of gastric, colon, liver, and lung tumors and abnormal crypt foci in mice and rats [18]. Despite its high salt content, miso was reported to prevent the induction of hypertension in Dahl salt-sensitive hypertensive rats [17]. Previous studies have revealed that miso intake has a protective effect against hypertension in Japanese individuals without hypertension [19,20]. In addition, habitual miso soup consumption is associated with lower insulin resistance [21,22]. Moreover, a recent study from Japan revealed that the intake of fermented soy products including miso, rather than soy products, is related to a lower risk of total mortality [23]. Furthermore, miso intake suppresses visceral fat accumulation and hepatic lipid deposits in mice [24]. Soybeans, which are the main ingredient of miso, contain bioactive substances such as isoflavones and dietary fiber, and are useful in the prevention of various diseases [25,26,27]. In addition, fermented soybean products contain more bioactive ingredients and nutrients than nonfermented soybean products [28,29]. Therefore, nutrients and bioactive components in fermented soybean products may have multifaceted benefits for health, including survival. However, no previous studies investigated the association between habitual miso consumption and sarcopenia in patients with T2DM. Furthermore, the effect of sex of the patient on this association is also an important component that has not been clarified yet. Therefore, this cross-sectional study researched the relationship between habitual miso consumption and sarcopenia in men and women with T2DM, separately.

## 2. Materials and Methods

### 2.1. Study Participants

The KAMOGAWA-DM cohort study, a prospective cohort study, started in 2014 and is currently ongoing [30]. In this cross-sectional study, patients with data of questionnaires from January 2016 to December 2018 were included. Exclusion criteria were as follows: no data of multifrequency impedance body composition analyzer, no data of hand grip strength, incomplete questionnaires, and patients without T2DM. This study was approved by the local research ethics committee (No. RBMR-E-466-6) and was carried out in accordance with the Declaration of Helsinki. All patients gave written informed consent.

### 2.2. Data Collection

Duration of diabetes, family history of diabetes, smoking status, and exercise habit were asked by a standardized questionnaire. The patients were divided into smokers and nonsmokers. We defined regular exercisers as patients who regularly played any type of sport >1×/week. Further, venous blood was collected from the participants who had fasted overnight, and the levels of triglycerides, high-density lipoprotein cholesterol, creatinine (Cr), uric acid, and fasting plasma glucose were measured. Estimated glomerular filtration rate (eGFR) was calculated using the equation of the Japanese Society of Nephrology, that is, eGFR = 194 × Cr-1.094 × age-0.287 (mL/min/1.73 m^2^) (×0.739, if woman) [31]. Hemoglobin A1c (HbA1c) levels were estimated using high-performance liquid chromatography and were expressed as National Glycohemoglobin Standardization Program (NGSP) units. After the participants had 5 min of rest, blood pressure was measured by a HEM-906 device (OMRON, Kyoto, Japan) in a quiet space, automatically. Additionally, the data on medications, including those for diabetes and hypertension (renin–angiotensin–aldosterone inhibitor) were obtained from the patients’ medical records.

### 2.3. Definition of Sarcopenia

Using InBody 720 (InBody, Japan, Tokyo, Japan), a multifrequency impedance body composition analyzer, the data of body weight (kg), appendicular muscle mass (kg), and body fat mass (kg) were gathered [32]. Then, body mass index (BMI, kg/m^2^), dividing body weight (kg) by the square of height (m) and ideal body weight (IBW), 22 multiplied by the square of the patient’s height in meters squared [33], were calculated. Percent body fat mass (%), dividing body fat mass (kg) by body weight (kg) × 100 and skeletal muscle mass index (SMI, %), appendicular muscle mass divided by body weight × 100 [34] for muscle mass were also calculated. Hand grip strength (HGS) was measured twice with each hand on a handgrip dynamometer (Smedley; Takei Scientific Instruments, Niigata, Japan), and the highest value was recorded. Adjusted grip strength (AGS) was calculated as HGS value divided by body weight × 100 [35] for muscle strength. The cutoff points for low muscle mass and low muscle strength were 28.64% and 51.26% in men and 24.12% and 35.38% in women [36]. Having both low muscle mass and low muscle strength was defined as sarcopenia [36].

### 2.4. Data of Habitual Diet Intake, Including Habitual Miso Consumption

Using a brief-type self-administered diet history questionnaire (BDHQ), habitual food and nutrient intake during the preceding 1 month period were evaluated [37]. The details and validity of BDHQ have been presented previously [38]. Patient data on the intakes of energy, fat, carbohydrate, protein, including animal and vegetable proteins, fiber, miso soup consumption, and alcohol consumption were obtained by BDHQ. Energy intake (kcal/kg IBW/day), protein intake (g/kg IBW/day), animal protein intake (g/kg IBW/day), vegetable protein intake (g/kg IBW/day), fat intake (g/kg IBW/day), and carbohydrate intake (g/kg IBW/day) were calculated. The ratio of carbohydrate to fiber intake was defined as carbohydrate intake divided by fiber intake [39]. In addition, the data on the intake frequency of miso soup were also collected and habitual miso consumption was defined as consuming miso soup regularly.

### 2.5. Statistical Analysis

The data are shown as means (standard deviation (SD)) or frequencies of potential confounding variables. Because the characteristics and dietary intakes differed between men and women, the data was analyzed by sex.

Patients were divided into two groups according to habitual miso consumption. We performed a Shapiro–Wilk test to investigate the distribution of variables. The differences in the continuous variables and categorical variables were evaluated by Student’s t-test or the Mann–Whitney U test and the chi-square test, respectively. The correlation was analyzed by Pearson’s correlation coefficient.

Further, to examine the effects of habitual miso consumption on the prevalence of sarcopenia, a logistic regression analysis was performed. The independent variables were sex, age, insulin treatment, habit of exercise, habit of smoking, duration of diabetes, HbA1c, energy intake, protein intake, and alcohol consumption. In addition, we performed a logistic regression analysis to examine the effect of miso soup intake, as a continuous variable, on the prevalence of sarcopenia. The independent variables were sex, age, insulin treatment, habit of exercise, habit of smoking, duration of diabetes, HbA1c, energy intake, protein intake, and alcohol consumption. Because miso soup intake was a skewed variable, logarithmic transformation was done before performing a logistic regression analysis, which were performed to evaluate the association of sarcopenia with log (miso soup intake + 1).

We performed subanalyses to examine the effects of habitual miso soup consumption on the prevalence of sarcopenia adjusted by age, according to the presence or absence of habit of smoking.

We used EZR (Saitama Medical Center, Jichi Medical University, Saitama, Japan) [40] for statistical analyses. Differences with *p* values < 0.05 were set as statistically significant.

## 3. Results

In the present study, 523 patients (276 men and 247 women) were included. Among them, 109 patients (50 men and 59 women) who did not undergo the bioelectrical impedance analysis (BIA) test, 24 patients with no date of BDHQ (15 men and 9 women), a patient who did not undergo the hand grip strength test, and 38 patients without T2DM (19 men and 19 women) were excluded from the study. Therefore, 351 patients (192 men and 159 women) were the study participants (Figure 1).

Table 1 shows the clinical characteristics of the study participants. Mean age was 66.6 ± 10.6 years in all subjects. Mean BMI was 23.9 ± 3.7 kg/m^2^ in men and 25.1 ± 5.1 kg/m^2^ in women. Mean SMI and AGS were 31.7 ± 3.2% and 50.7 ± 10.1% in men and 25.7 ± 2.9% and 36.1 ± 9.0% in women, respectively. The proportions of men and women with sarcopenia were 8.7% (*n* = 22/192) and 22.6% (*n* = 36/159), respectively, and the proportion of women with sarcopenia was significantly higher than that of men with sarcopenia (*p* = 0.008).

Table 2 shows the results of the data of dietary intake. The percentages of habitual miso consumption were 88.0% (*n* = 169/192) and 83.6% (*n* = 133/159) in men and women, respectively.

Table 3 shows the clinical characteristics of patients according to habitual miso consumption. Both in men (24.4 ± 7.2% vs. 27.4 ± 5.6%, *p* = 0.056) and women (33.8 ± 8.0% vs. 39.3 ± 7.3%, *p* = 0.001), patients with habitual miso consumption had a lower percentage of body fat than those without it. Among women, patients with habitual miso consumption had lower BMI (24.7 ± 5.0 kg/m^2^ vs. 27.3 ± 5.0 kg/m^2^, *p* = 0.022) and lower HbA1c levels (7.2 ± 1.0% vs. 8.1 ± 1.9%, *p* = 0.006) than those without it. In addition, in women with habitual miso consumption, the proportions of patients with low skeletal muscle mass (35/133 (26.3%) vs. 13/26 (50.0%), *p* = 0.024), low muscle strength (60/133 (45.1%) vs. 19/26 (73.1%), *p* = 0.009), and sarcopenia (25/133 (18.8%) vs. 11/26 (42.3%), *p* = 0.018) were lower than those in women without habitual miso consumption.

Table 4 shows the habitual diet intakes according to the presence of habitual miso consumption. In women, the patients with habitual miso consumption had higher intakes of energy (30.8 ± 11.2 kg/IBW kg/day vs. 23.5 ± 6.3 kg/IBW kg/day, *p* < 0.001), protein (1.4 ± 0.6 g/IBW kg/day vs. 1.0 ± 0.3 g/IBW kg/day, *p* < 0.001), animal protein (0.9 ± 0.5 g/IBW kg/day vs. 0.6 ± 0.3 g/IBW kg/day, *p* = 0.006), vegetable protein (0.5 ± 0.2 g/IBW kg/day vs. 0.4 ± 0.1 g/IBW kg/day, *p* < 0.001), fat (1.0 ± 0.5 g/IBW kg/day vs. 0.8 ± 0.3 g/IBW kg/day, *p* = 0.006), carbohydrate (3.9 ± 1.4 g/IBW kg, day vs. 3.0 ± 0.8 g/IBW kg/day, *p* = 0.003), and dietary fiber (12.4 ± 4.6 g/day vs. 8.1 ± 2.9 g/day, *p* < 0.001) and lower carbohydrate-to-fiber ratios (17.0 ± 5.6 vs. 20.1 ± 6.6, *p* = 0.028) in their diets than those without habitual miso consumption.

Figure 2 shows the correlation between the log (miso soup intake + 1) and skeletal muscle mass or muscle strength. In women, the log (miso soup intake + 1) was correlated with skeletal muscle mass index (*r* = 0.25, *p* = 0.003) and adjusted grip strength (*r* = 0.17, *p* = 0.030). However, the log (miso soup intake + 1) was not correlated with skeletal muscle mass index (*r* = 0.14, *p* = 0.079) and adjusted grip strength (*r* = 0.14, *p* = 0.061) in men.

Furthermore, habitual miso consumption was negatively related to the presence of sarcopenia (adjusted OR, 0.20 (95% CI: 0.06–0.62), *p* = 0.005) in women, whereas this association was absent in men (adjusted OR, 1.11 (95% CI: 0.27–4.57), *p* = 0.882) (Table 5).

The log (miso soup intake + 1) was negatively related to the presence of sarcopenia (adjusted OR, 0.67 (95% CI: 0.52–0.86), *p* = 0.002) in women, whereas this association was absent in men (adjusted OR, 1.01 (95% CI: 0.76–1.34), *p* = 0.952) (Table 6).

In men, the association between habitual miso consumption and the presence of sarcopenia was absent in men both with (adjusted OR, 0.68 (95% CI: 0.02–29.90), *p* = 0.839), and without a habit of smoking (adjusted OR, 1.26 (95% CI: 0.27–5.98), *p* = 0.768). On the other hand, in women without a habit of smoking, habitual miso consumption was negatively related to the presence of sarcopenia (adjusted OR, 0.36 (95% CI: 0.14–0.94), *p* = 0.036), whereas habitual miso consumption was not associated with the presence of sarcopenia in women with a habit of smoking (adjusted OR, 0.07 (95% CI: 0.00–3.70), *p* = 0.187).

## 4. Discussion

We firstly investigated the relationship between habitual miso consumption and the presence of sarcopenia in patients with T2DM in this study. We revealed that habitual miso consumption was associated with a low prevalence of sarcopenia in women but not in men. Additionally, habitual miso consumption was associated with lower percentage of body fat mass in women.

The possible explanation for the association between habitual miso consumption and the low prevalence of sarcopenia is as follows. A previous study has shown that miso consumption suppresses fat accumulation in fat mass and showed anti-obesity effects [24]. Furthermore, there is a close association between increased visceral fat and muscle atrophy [41,42]. Secretion of proinflammatory cytokines such as interleukin-6 and tumor necrosis factor-α from obesity-induced hypertrophic fat cells causes muscle atrophy [43]. The consumption of miso soup every day is related to lower insulin resistance in women [21]. In fact, the percent body fat mass was lower in women with habitual miso consumption than in those without it, in this study. Furthermore, miso is a product made of fermented soybeans. Soybeans are rich in soybean-specific proteins (glycinin and conglycinin), vitamin E, isoflavones (daidzein, daidzin, genistein, and glycitein), lipids rich in polyunsaturated fatty acids, lecithin, and saponin [17,44,45]. Miso also contains pyroglutamyl leucine [46], which is spontaneously generated from peptides with a glutaminyl residue at the amino terminal, during storage and processing [47]. Pyroglutamyl leucine was found to improve gut dysbiosis and colitis in mice [46,48,49] and, in fact, suppresses the excess proliferation of phylum *Firmicutes*, which is associated with obesity [49]. The main ingredient of miso is soybeans. Soybeans contain high amounts of omega-3 and omega-6 fatty acids [50]. There is a negative correlation between omega-3 fatty acids intake and the presence of sarcopenia in elderly patients with T2DM [51]. In addition, both omega-3 and omega-6 intake is known to be important for muscle maintenance [52]. Moreover, soybean proteins and isoflavones are known to reduce accumulation of visceral fat mass in animal models [53,54]. Isoflavones have a structure similar to that of estradiol and have a high binding ability to the primary estrogen receptor in the vascular wall [55]. Therefore, isoflavones in miso may inhibit the accumulation of visceral fat and the loss of muscle mass and strength in women.

Another possible mechanism is the association between habitual miso consumption and diet quality. In this study, women without habitual miso consumption had lower energy intakes than those with habitual miso consumption. Moreover, compared to those with habitual miso consumption, the diets of women without habitual miso consumption showed higher carbohydrate-to-fiber ratios, which has a reported association with metabolic syndrome [39] and visceral fat accumulation [56]. Thus, women without habitual miso consumption may have had poor diet qualities, which may have contributed to obesity and sarcopenia. Adherence to a dietary pattern rich in foods characteristic of the Japanese diet is associated with a lower prevalence of sarcopenia [57]. Miso soup is well-known as a traditional Japanese food. Therefore, the consumption of miso soup contributes to the traditional Japanese diet, suggesting that women with habitual miso consumption have a lower prevalence of sarcopenia. On the other hand, habitual miso consumption was not associated with the presence of sarcopenia in men. However, it is unclear why there is no association between miso consumption and sarcopenia in men. It is probably because there was no association between habitual miso consumption and diet quality in men. In fact, a previous study revealed an association between habitual miso soup consumption and insulin resistance in women but not in men [21].

In women without a habit of smoking, habitual miso consumption was associated with the presence of sarcopenia, but in women with a habit of smoking, habitual miso consumption was not associated with the presence of sarcopenia. This is due to the high oxidative stress caused by smoking [58]. In fact, the habit of smoking is known to be a risk factor of sarcopenia [59].

However, our study had certain limitations. First, the data regarding the frequency of miso soup consumption were self-reported, and there were some concerns regarding the accuracy of the data. Second, because of a cross-sectional study, it was not possible to show the causal relationship. Third, the sample size of this study was relatively small. Fourth, women without habitual miso consumption might have poor quality of diet, thus, we cannot deny the possibility that diet quality contributed to this result. Finally, all study participants were Japanese patients only; hence, it is unclear whether the results can be generalized to other ethnic groups.

In conclusion, this study firstly revealed that habitual miso consumption was associated with a low presence of sarcopenia in women but not in men. Although further research is needed, miso soup is a part of the traditional Japanese diet, and it has been suggested that consuming miso soup may help prevent sarcopenia.

## Figures and Tables

**Figure 1 nutrients-13-00072-f001:**
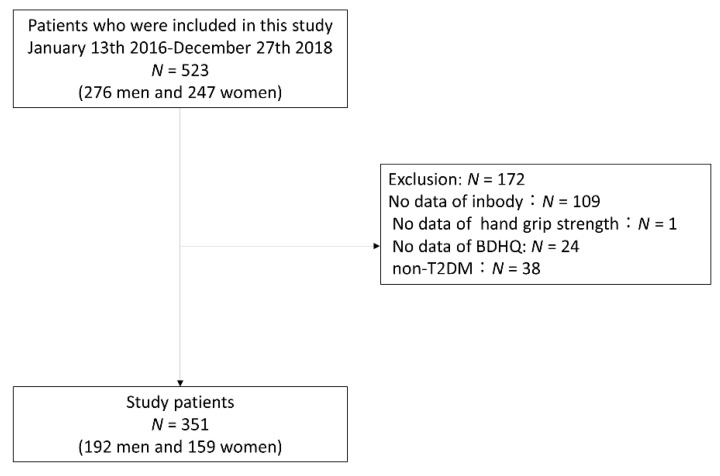
Inclusion and exclusion flow. BDHQ, Brief-type self-administered diet history questionnaire; T2DM, type 2 diabetes mellitus.

**Figure 2 nutrients-13-00072-f002:**
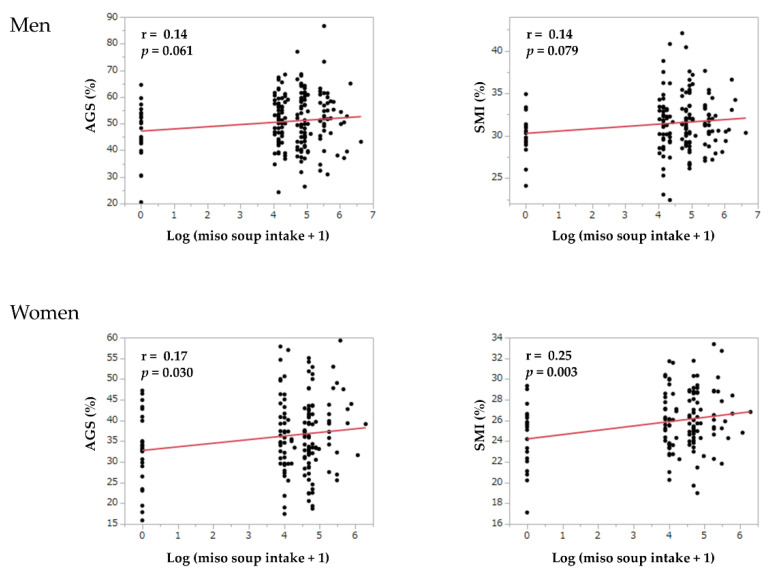
The correlation between miso soup intake and skeletal muscle mass or muscle strength. AGS, adjusted grip strength; log, logarithm; SMI, skeletal muscle mass index.

**Table 1 nutrients-13-00072-t001:** Clinical characteristics of study participants.

	All *n* = 351	Men *n* = 192	Women *n* = 159	*p*
Age, years	66.6 (10.6)	67.1 (10.9)	66.0 (10.4)	0.430
Duration of diabetes, years	14.1 (10.0)	14.8 (9.5)	13.3 (10.6)	0.041
Family history of diabetes (−/+)	194/157	113/79	81/78	0.169
Height, cm	160.9 (9.2)	167.2 (6.3)	153.2 (5.6)	<0.001
Body weight, kg	62.9 (12.2)	66.9 (11.4)	58.0 (11.4)	<0.001
Body mass index, kg/m^2^	24.5 (4.4)	23.9 (3.7)	25.1 (5.1)	0.062
SBP, mmHg	134.4 (19.0)	133.8 (18.3)	135.2 (20.0)	0.690
DBP, mmHg	78.9 (11.3)	79.3 (10.9)	78.5 (11.8)	0.198
Insulin (−/+)	263/88	145/47	118/41	0.875
DPP4 inhibitor (−/+)	169/182	86/106	83/76	0.202
SU (−/+)	270/81	144/48	126/33	0.417
Glinide (−/+)	320/31	180/12	140/19	0.092
Biguanide (−/+)	207/144	118/74	89/70	0.352
Thiazolidinediones (−/+)	338/13	183/9	155/4	0.430
α- GI (−/+)	306/45	169/23	137/22	0.721
SGLT2 inhibitor (−/+)	294/57	160/32	134/25	0.926
GLP-1 receptor agonist (−/+)	296/55	167/25	129/30	0.176
RAS inhibitor (−/+)	194/157	98/94	96/63	0.100
Smoking (−/+)	301/50	152/40	149/10	<0.001
Habit of exercise (−/+)	181/170	95/97	86/73	0.452
HbA1c, mmol/mol	56.8 (13.6)	57.1 (13.6)	56.4 (13.6)	0.777
HbA1c, %	7.3 (1.2)	7.4 (1.2)	7.3 (1.2)	0.777
Plasma glucose, mmol/L	8.3 (2.8)	8.4 (2.7)	8.1 (2.8)	0.130
Creatinine, umol/L	73.2 (32.1)	83.4 (36.2)	60.9 (20.5)	<0.001
eGFR, mL/min/1.73 m^2^	69.7 (19.3)	69.1 (20.3)	70.5 (18.1)	0.524
Uric acid, mmol/L	305.6 (74.4)	319.3 (78.4)	288.8 (65.8)	<0.001
Triglycerides, mmol/L	1.5 (0.9)	1.5 (0.9)	1.5 (0.8)	0.854
HDL cholesterol, mmol/L	1.6 (0.4)	1.5 (0.4)	1.6 (0.4)	<0.001
Body fat mass, kg	19.0 (8.5)	17.2 (7.6)	21.3 (9.1)	<0.001
Percent body fat mass, %	29.3 (9.0)	24.8 (7.1)	34.7 (8.1)	<0.001
Appendicular muscle mass, kg	18.3 (4.2)	21.1 (3.2)	14.9 (2.4)	<0.001
SMI, %	29.0 (4.3)	31.7 (3.2)	25.7 (2.9)	<0.001
Low skeletal muscle mass (−/+)	275/76	164/28	111/48	0.001
HGS, kg	27.9 (9.3)	33.7 (7.7)	20.8 (5.2)	<0.001
AGS, %	44.1 (12.0)	50.7 (10.1)	36.1 (9.0)	<0.001
Low muscle strength (−/+)	175/176	95/97	80/79	0.961
Presence of sarcopenia (−/+)	293/58	170/22	123/36	0.008

Data was expressed as mean (standard deviation) or number. The difference between groups was evaluated by Student’s *t*-test, Mann–Whitney U test or chi-square test. α-GI, alpha-glucosidase inhibitor; AGS; adjusted grip strength; DBP, diastolic blood pressure; eGFR, estimated glomerular filtration rate; HDL, high-density lipoprotein; HGS, handgrip strength, RAS, renin–angiotensin system; SBP, systolic blood pressure; SMI, skeletal muscle mass index.

**Table 2 nutrients-13-00072-t002:** Habitual diet intake of study participants.

	All *n* = 351	Men *n* = 192	Women *n* = 159	*p*
Total energy intake, kcal/day	1748.4 (636.9)	1933.0 (646.4)	1525.4 (549.6)	<0.001
Energy intake, kcal/IBW kg/day	30.7 (10.9)	31.5 (10.8)	29.6 (10.9)	0.058
Total fat intake, g/day	55.4 (23.4)	59.0 (23.4)	51.1 (22.9)	<0.001
Fat intake, g/IBW kg/day	1.0 (0.4)	1.0 (0.4)	1.0 (0.5)	0.836
Fat intake per energy intake, %	28.7 (6.3)	27.6 (6.5)	30.0 (5.9)	<0.001
Total protein intake, g/day	73.0 (29.7)	76.8 (30.0)	68.4 (28.8)	0.002
Protein intake, g/IBW kg/day	1.3 (0.5)	1.3 (0.5)	1.3 (0.6)	0.220
Protein intake per energy intake, %	16.8 (3.5)	16.0 (3.1)	17.9 (3.5)	<0.001
Animal protein intake, g/day	44.8 (23.1)	46.4 (23.4)	42.8 (22.7)	0.072
Animal protein intake, g/IBW kg/day	0.8 (0.4)	0.8 (0.4)	0.8 (0.5)	0.105
Vegetable protein intake, g/day	28.2 (10.0)	30.4 (10.3)	25.6 (9.0)	<0.001
Vegetable protein intake, g/IBW kg/day	0.5 (0.2)	0.5 (0.2)	0.5 (0.2)	0.874
Total carbohydrate intake, g/day	220.3 (86.1)	243.0 (90.5)	193.0 (71.7)	<0.001
Carbohydrate intake, g/IBW kg/day	3.9 (1.5)	4.0 (1.5)	3.7 (1.4)	0.125
Carbohydrate intake per energy intake, %	50.6 (8.7)	50.5 (9.1)	50.8 (8.2)	0.708
Dietary fiber intake, g/day	12.2 (5.1)	12.6 (5.4)	11.7 (4.7)	0.209
Carbohydrate/fiber ratio	19.5 (7.3)	21.1 (7.9)	17.5 (5.9)	<0.001
Alcohol consumption, g/day	8.0 (20.0)	14.1 (25.3)	0.7 (3.0)	<0.001
Miso soup intake, g/day	115.9 (103.0)	132.8 (114.5)	95.6 (83.1)	<0.001
Habitual miso consumption (−/+)	49/302	23/169	26/133	0.307

Data was expressed as mean (standard deviation) or number. The difference between groups was evaluated by Student’s *t*-test, Mann–Whitney U test or chi-square test. IBW, ideal body weight.

**Table 3 nutrients-13-00072-t003:** Clinical characteristics according to habitual miso consumption.

	Men, *n* = 192			Women, *n* = 159		
	Habitual Miso Consumption (−) *n* = 23	Habitual Miso Consumption (+) *n* = 169	*p*	Habitual Miso Consumption (−) *n* = 26	Habitual Miso Consumption (+) *n* = 133	*p*
Age, years	70.8 (10.6)	66.6 (10.8)	0.133	64.3 (13.4)	66.3 (9.7)	0.699
Duration of diabetes, years	18.9 (9.4)	14.2 (9.4)	0.014	17.8 (12.5)	12.5 (10.0)	0.018
Family history of diabetes (−/+)	14/9	99/70	1.000	10/16	71/62	0.239
Height, cm	165.7 (7.0)	167.4 (6.1)	0.227	152.3 (6.2)	153.4 (5.5)	0.382
Body weight, kg	66.5 (10.6)	66.9 (11.5)	0.906	62.5 (11.5)	57.2 (11.2)	0.022
Body mass index, kg/m^2^	24.2 (2.9)	23.9 (3.8)	0.567	27.3 (5.3)	24.7 (5.0)	0.022
SBP, mmHg	127.4 (16.6)	134.6 (18.4)	0.087	138.2 (18.6)	134.6 (20.2)	0.215
DBP, mmHg	74.8 (10.8)	79.9 (10.8)	0.034	76.0 (9.0)	79.0 (12.2)	0.253
Insulin (−/+)	20/3	125/44	0.271	15/11	103/30	0.063
DPP4 (−/+)	9/14	77/92	0.720	12/14	71/62	0.645
SU (−/+)	13/10	131/38	0.054	19/7	107/26	0.559
Glinide (−/+)	21/2	159/10	0.954	23/3	117/16	1.000
Biguanide (−/+)	9/14	109/60	0.034	16/10	73/60	0.683
Thiazolidinediones (−/+)	22/1	161/8	1.000	25/1	130/3	1.000
α- GI (−/+)	18/5	151/18	0.232	24/2	113/20	0.496
SGLT2 (−/+)	16/7	144/25	0.112	23/3	111/22	0.729
GLP-1 (−/+)	18/5	149/20	0.320	18/8	111/22	0.155
RAS inhibitor (−/+)	10/13	88/81	0.582	12/14	84/49	0.161
Smoking (−/+)	19/4	133/36	0.873	23/3	126/7	0.445
Habit of exercise (−/+)	14/9	81/88	0.346	14/12	72/61	1.000
HbA1c, mmol/mol	58.2 (17.1)	56.9 (13.1)	0.908	65.3 (20.3)	54.7 (11.1)	0.006
HbA1c, %	7.5 (1.6)	7.4 (1.2)	0.908	8.1 (1.9)	7.2 (1.0)	0.006
Plasma glucose, mmol/L	8.2 (2.7)	8.4 (2.7)	0.593	9.1 (3.9)	7.9 (2.5)	0.084
Creatinine, umol/L	81.9 (17.4)	83.6 (38.1)	0.305	68.0 (35.1)	59.5 (16.0)	0.994
eGFR, mL/min/1.73 m^2^	66.0 (19.5)	69.6 (20.4)	0.435	69.3 (23.7)	70.7 (16.9)	0.726
Uric acid, mmol/L	323.0 (59.9)	318.8 (80.7)	0.843	296.5 (67.8)	287.3 (65.6)	0.585
Triglycerides, mmol/L	1.4 (0.7)	1.5 (0.9)	0.876	1.5 (0.7)	1.4 (0.9)	0.453
HDL cholesterol, mmol/L	1.5 (0.4)	1.5 (0.5)	0.469	1.6 (0.3)	1.6 (0.4)	0.567
Body fat mass, kg	18.6 (6.1)	17.0 (7.8)	0.189	25.6 (9.6)	20.4 (8.7)	0.009
Percent body fat mass, %	27.4 (5.6)	24.4 (7.2)	0.056	39.3 (7.3)	33.8 (8.0)	0.001
Appendicular muscle mass, kg	20.4 (3.3)	21.2 (3.1)	0.233	15.2 (2.7)	14.9 (2.3)	0.507
SMI, %	30.7 (2.3)	31.9 (3.3)	0.078	24.3 (3.0)	26.0 (2.8)	0.011
Low skeletal muscle mass (−/+)	19/4	145/25	0.735	14/13	98/35	0.024
HGS, kg	31.0 (7.1)	34.0 (7.8)	0.138	20.4 (5.2)	20.9 (5.2)	0.872
AGS, %	46.9 (10.3)	51.2 (10.0)	0.055	32.9 (8.6)	36.7 (8.9)	0.043
Low muscle strength (−/+)	9/14	87/83	0.278	7/19	73/60	0.009
Presence of sarcopenia (−/+)	20/3	150/19	1.000	15/11	108/25	0.018

Data was expressed as mean (standard deviation) or number. The difference between groups was evaluated by Student’s *t*-test, Mann–Whitney U test or chi-square test. α-GI, alpha-glucosidase inhibitor; AGS; adjusted grip strength; DBP, diastolic blood pressure; HDL, high-density lipoprotein; HGS, handgrip strength; eGFR, estimated glomerular filtration rate; RAS, renin-angiotensin system; SBP, systolic blood pressure; SMI, skeletal muscle mass index.

**Table 4 nutrients-13-00072-t004:** Habitual diet intake according to habitual miso consumption.

	Men, *n* = 192			Women, *n* = 159		
	Habitual Miso Consumption (−) *n* = 23	Habitual Miso Consumption (+) *n* = 169	*p*	Habitual Miso Consumption (−) *n* = 26	Habitual Miso Consumption (+) *n* = 133	*p*
Total energy intake, kcal/day	1767.8 (566.0)	1955.5 (654.8)	0.099	1201.1 (333.9)	1588.8 (561.9)	<0.001
Energy intake, kcal/IBW kg/day	29.4 (10.2)	31.8 (10.9)	0.192	23.5 (6.3)	30.8 (11.2)	<0.001
Total protein intake, g/day	71.5 (23.4)	77.5 (30.7)	0.383	51.2 (18.1)	71.7 (29.3)	<0.001
Protein intake, g/IBW kg/day	1.2 (0.4)	1.3 (0.5)	0.469	1.0 (0.3)	1.4 (0.6)	<0.001
Protein intake per energy intake, %	16.3 (3.2)	15.9 (3.1)	0.497	17.1 (3.9)	18.0 (3.4)	0.102
Animal protein intake, g/day	44.3 (19.1)	46.7 (24.0)	0.672	32.2 (14.5)	44.8 (23.5)	0.005
Animal protein intake, g/IBW kg/day	0.7 (0.3)	0.8 (0.4)	0.817	0.6 (0.3)	0.9 (0.5)	0.006
Vegetable protein intake, g/day	27.2 (9.3)	30.8 (10.4)	0.051	19.0 (6.0)	26.9 (9.0)	<0.001
Vegetable protein intake, g/IBW kg/day	0.5 (0.2)	0.5 (0.2)	0.073	0.4 (0.1)	0.5 (0.2)	<0.001
Total fat intake, g/day	61.9 (19.0)	58.6 (23.9)	0.295	40.3 (15.1)	53.3 (23.5)	0.004
Fat intake, g/IBW kg/day	1.0 (0.3)	1.0 (0.4)	0.256	0.8 (0.3)	1.0 (0.5)	0.006
Fat intake per energy intake, %	32.1 (6.8)	27.0 (6.2)	0.001	29.9 (7.0)	30.0 (5.7)	0.939
Total carbohydrate intake, g/day	218.8 (100.7)	246.2 (88.8)	0.031	154.0 (44.2)	200.6 (73.6)	0.002
Carbohydrate intake, g/IBW kg/day	3.7 (1.8)	4.0 (1.5)	0.060	3.0 (0.8)	3.9 (1.4)	0.003
Carbohydrate intake per energy intake, %	49.2 (9.9)	50.6 (9.0)	0.468	61.6 (9.5)	50.7 (8.0)	0.584
Dietary fiber intake, g/day	11.1 (3.6)	12.8 (5.6)	0.302	8.1 (2.9)	12.4 (4.6)	<0.001
Carbohydrate/fiber ratio	21.1 (8.4)	21.1 (7.8)	0.914	20.1 (6.6)	17.0 (5.6)	0.028
Alcohol consumption, g/day	4.1 (11.9)	15.5 (26.3)	0.003	0.4 (1.5)	0.7 (3.2)	0.800

Data was expressed as mean (standard deviation) or number. The difference between groups was evaluated by Student’s *t*-test, Mann–Whitney U test or chi-square test, IBW, ideal body weight.

**Table 5 nutrients-13-00072-t005:** Odds ratio of habitual miso consumption on the presence of sarcopenia.

**Men**	**Model 1**		**Model 2**		**Model 3**	
	**OR (95% CI)**	***p***	**OR (95% CI)**	***p***	**OR (95% CI)**	***p***
Habitual miso consumption (+)	0.84 (0.23–3.11)	0.799	0.91 (0.24–3.40)	0.887	1.11 (0.27–4.57)	0.882
Age (years)	-	-	1.02 (0.97–1.06)	0.425	1.03 (0.97–1.08)	0.361
HbA1c (mmol/mol)	-	-	-	-	0.99 (0.95–1.04)	0.789
Insulin treatment	-	-	-	-	0.30 (0.06–1.46)	0.137
Habit of exercise	-	-	-	-	0.76 (0.29–1.95)	0.564
Habit of smoking	-	-	-	-	0.63 (0.17–2.33)	0.484
Duration of diabetes (years)	-	-	-	-	0.98 (0.92–1.04)	0.456
Energy intake (kcal/IBW kg/day)	-	-	-	-	0.96 (0.87–1.06)	0.424
Protein intake (g/IBW kg/day)	-	-	-	-	1.24 (0.17–9.09)	0.831
Alcohol consumption (g/day)	-	-	-	-	0.99 (0.97–1.02)	0.614
**Women**	**Model 1**		**Model 2**		**Model 3**	
	**OR (95% CI)**	***p***	**OR (95% CI)**	***p***	**OR (95% CI)**	***p***
Habitual miso consumption (+)	0.32 (0.13–0.77)	0.011	0.33 (0.13–0.80)	0.015	0.20 (0.06–0.62)	0.005
Age (years)	-	-	0.98 (0.94–1.01)	0.212	1.01 (0.96–1.05)	0.795
HbA1c (mmol/mol)	-	-	-	-	1.01 (0.98–1.04)	0.692
Insulin treatment	-	-	-	-	0.64 (0.22–1.88)	0.419
Habit of exercise	-	-	-	-	0.79 (0.34–1.84)	0.579
Habit of smoking	-	-	-	-	2.07 (0.49–8.71)	0.321
Duration of diabetes (years)	-	-	-	-	0.95 (0.91–1.01)	0.085
Energy intake (kcal/IBW kg/day)	-	-	-	-	1.04 (0.96–1.13)	0.296
Protein intake (g/IBW kg/day)	-	-	-	-	0.65 (0.14–3.05)	0.584
Alcohol consumption (g/day)	-	-	-	-	1.07 (0.95–1.20)	0.298

Model 1 is unadjusted; Model 2 is adjusted for sex, age; Model 3 is adjusted for sex, age, duration of diabetes, HbA1c, habit of exercise, habit of smoking, insulin treatment, energy intake, protein intake, and alcohol consumption, IBW, ideal body weight.

**Table 6 nutrients-13-00072-t006:** Odds ratio of the quantity of miso soup intake on the presence of sarcopenia.

**Men**	**Model 1**		**Model 2**		**Model 3**	
	**OR (95% CI)**	***p***	**OR (95% CI)**	***p***	**OR (95% CI)**	***p***
Log (miso soup intake + 1)	0.94 (0.73–1.20)	0.618	0.95 (0.74–1.23)	0.703	1.01 (0.76–1.34)	0.952
Age (years)	-	-	1.02 (0.97–1.06)	0.445	1.03 (0.97–1.08)	0.368
HbA1c (mmol/mol)	-	-	-	-	0.94 (0.60–1.48)	0.783
Insulin treatment	-	-	-	-	0.31 (0.06–1.48)	0.141
Habit of exercise	-	-	-	-	0.76 (0.30–1.96)	0.574
Habit of smoking	-	-	-	-	0.63 (0.17–2.33)	0.484
Duration of diabetes (years)	-	-	-	-	0.98 (0.92–1.04)	0.448
Energy intake (kcal/IBW kg/day)	-	-	-	-	0.96 (0.87–1.06)	0.418
Protein intake (g/IBW kg/day)	-	-	-	-	1.25 (0.17–9.20)	0.823
Alcohol consumption (g/day)	-	-	-	-	0.99 (0.97–1.02)	0.621
**Women**	**Model 1**		**Model 2**		**Model 3**	
	**OR (95% CI)**	***p***	**OR (95% CI)**	***p***	**OR (95% CI)**	***p***
Log (miso soup intake + 1)	0.77 (0.64–0.93)	0.007	0.77 (0.64–0.94)	0.008	0.67 (0.52–0.86)	0.002
Age (years)	-	-	0.98 (0.94–1.01)	0.192	1.00 (0.96–1.05)	0.832
HbA1c (mmol/mol)	-	-	-	-	1.04 (0.75–1.45)	0.825
Insulin treatment	-	-	-	-	0.66 (0.22–1.96)	0.457
Habit of exercise	-	-	-	-	0.76 (0.32–1.81)	0.541
Habit of smoking	-	-	-	-	2.28 (0.54–9.69)	0.262
Duration of diabetes (years)	-	-	-	-	0.95 (0.90–1.00)	0.067
Energy intake (kcal/IBW kg/day)	-	-	-	-	1.05 (0.97–1.14)	0.261
Protein intake (g/IBW kg/day)	-	-	-	-	0.69 (0.14–3.30)	0.639
Alcohol consumption (g/day)	-	-	-	-	1.07 (0.95–1.21)	0.262

Model 1 is unadjusted; Model 2 is adjusted for sex, age; Model 3 is adjusted for sex, age, duration of diabetes, HbA1c, habit of exercise, habit of smoking, insulin treatment, energy intake, protein intake, and alcohol consumption, IBW, ideal body weight.

## Data Availability

The data presented in this study are available on request from the corresponding author. The data are not publicly available due to restrictions eg privacy or ethical.

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
