# Peer review of "Habitual Miso (Fermented Soybean Paste) Consumption Is Associated with a Low Prevalence of Sarcopenia in Patients with Type 2 Diabetes: A Cross-Sectional Study"

_nutrients, 2020, doi:10.3390/nu13010072_

Round 1

Reviewer 1 Report

In this study, the authors investigate if miso consumption is associated with sarcopenia in a cohort of 351 Japanese adults with T2D. They conclude that regular miso consumption is associated with a lower prevalence of sarcopenia in women. However, I do not believe that the authors can make this conclusion based on this study. In total, only 36 women had sarcopenia: 25 consumed miso and 11 did not consume miso. This sample size is insufficient to make a health claim. Aside from this, the mechanisms proposed in Lines 191-205 seem speculative. It seems more likely that women who do not consume miso regularly tend to have poorer diets, as discussed in Lines 206-207.

Minor:

Lines 156-163: Please include the n number along with the p-values. The percentages are misleading.

Author Response

Point 1

In this study, the authors investigate if miso consumption is associated with sarcopenia in a cohort of 351 Japanese adults with T2D. They conclude that regular miso consumption is associated with a lower prevalence of sarcopenia in women. However, I do not believe that the authors can make this conclusion based on this study. In total, only 36 women had sarcopenia: 25 consumed miso and 11 did not consume miso. This sample size is insufficient to make a health claim.

Response

Thank you for your comment. As you say, the sample size might be insufficient to make a conclusion. Thus, we have mentioned this point as one of the limitations of this study in the Discussion section and have changed the Conclusion section described as below.

Discussion (Line 278-279)

“Third, the sample size of this study was relatively small.”

Conclusion (Line 286-287)

“Although further research is needed, miso soup is a part of the traditional Japanese diet, and it has been suggested that consuming miso soup may help prevent sarcopenia.”

On the other hand, we have additionally investigated the effect of the miso soup intake, as a continuous variable, on sarcopenia.

In women, the miso soup intake was correlated with skeletal muscle mass index (r = 0.25, p = 0.003) and adjusted grip strength (r = 0.17, p = 0.030). However, the miso soup intake was not correlated with skeletal muscle mass index (r = 0.14, p = 0.079) and adjusted grip strength (r = 0.14, p = 0.061) in men. Moreover, miso soup intake was negatively related to the presence of sarcopenia (adjusted OR, 0.67 [95% CI: 0.52–0.86], p = 0.002) in women, whereas this association was absent in men (adjusted OR, 1.01 [95% CI: 0.76–1.34], p = 0.952). Thus, there is an association between miso soup intake and the presence of sarcopenia. We have added these points in the Materials and Methods and the Results sections as below.

Materials and Methods (Line 122-123)

“The correlation was analyzed by Pearson’s correlation coefficient.”

Materials and Methods (Line 127-132)

“In addition, we performed a logistic regression analysis to examine the effect of miso soup intake, as a continuous variable, on the prevalence of sarcopenia. The independent variables were sex, age, insulin treatment, habit of exercise, habit of smoking, duration of diabetes, HbA1c, energy intake, protein intake, and alcohol consumption. Because the miso soup intake was a skewed variable, logarithmic transformation was done before performing a logistic regression analysis, which were performed to evaluate the association of sarcopenia with log (miso soup intake + 1).”

Results (Line 196-200)

“Figure 2 shows the correlation between the log (miso soup intake + 1) and skeletal muscle mass or muscle strength. In women, the log (miso soup intake + 1) was correlated with skeletal muscle mass index (r = 0.25, p = 0.003) and adjusted grip strength (r = 0.17, p = 0.030). However, the log (miso soup intake + 1) was not correlated with skeletal muscle mass index (r = 0.14, p = 0.079) and adjusted grip strength (r = 0.14, p = 0.061) in men.”

Results (Line 211-213)

“The log (miso soup intake + 1) was negatively related to the presence of sarcopenia (adjusted OR, 0.67 [95% CI: 0.52–0.86], p = 0.002) in women, whereas this association was absent in men (adjusted OR, 1.01 [95% CI: 0.76–1.34], p = 0.952) (Table 6).”

Point 2

Aside from this, the mechanisms proposed in Lines 191-205 seem speculative.

Response

Thank you for your valuable comment. As you say, some of the mechanisms might be speculative. However, in clinical practice, habitual miso soup consumption was associated with lower insulin resistance, and miso intake was also shown to have anti-inflammation effect in animal models. In addition, we have added the other mechanism of miso on the health. Miso contains pyroglutamyl leucine, which is spontaneously generated from peptides with a glutaminyl residue at the amino terminal, during storage and processing. Pyroglutamyl leucine was found to improve gut dysbiosis; and in fact, suppressed the excess proliferation of phylum Firmicutes, which is associated with obesity. According to your comment, we have modified the Discussion section described as below.

Discussion (Line 242-246)

“Miso also contains pyroglutamyl leucine [46], which is spontaneously generated from peptides with a glutaminyl residue at the amino terminal, during storage and processing [47]. Pyroglutamyl leucine was found to improve gut dysbiosis and colitis in mice [46,48,49] and in fact, suppressed the excess proliferation of phylum Firmicutes, which is associated with obesity [49].”

References

  1. Shirako, S.; Kojima, Y.; Tomari, N.; Nakamura, Y.; Matsumura, Y.; Ikeda, K.; Inagaki, N.; Sato, K. Pyroglutamyl leucine, a peptide in fermented foods, attenuates dysbiosis by increasing host antimicrobial peptide. NPJ science of food 2019, 3, 18. doi:10.1038/s41538-019-0050-z. [B] Ejima, A., Nakamura, M., Suzuki, Y. A., Sato, K. Identification of food-derived peptides in human blood after ingestion of corn and wheat gluten hydrolysates. J Food Bioact 2018; 2: 104-111.
  2. Ejima, A.; Nakamura, M.; Suzuki, Y. A.; Sato, K. Identification of food-derived peptides in human blood after ingestion of corn and wheat gluten hydrolysates. J Food Bioact 2018, 2, 104-111.
  3. Wada, S.; Sato, K.; Ohta, R.; Wada, E.; Bou, Y.; Fujiwara, M.; Kiyono, T.; Park, E. Y.; Aoi, W.; Takagi, T.; Naito, Y.; Yoshikawa, T. Ingestion of low dose pyroglutamyl leucine improves dextran sulfate sodium-induced colitis and intestinal microbiota in mice. J Agric Food Chem 2013, 61(37), 8807-8813. doi: 10.1021/jf402515a.
  4. Castaner, O.; Goday, A.; Park, Y.M.; Lee, S.H.; Magkos, F.; Toh Ee Shiow, S. A. The Gut Microbiome Profile in Obesity: A Systematic Review. Int J Endocrinol 2018; 2018: 4095789. doi:10.1155/2018/4095789.

Point 3

It seems more likely that women who do not consume miso regularly tend to have poorer diets, as discussed in Lines 206-207.

Response

Thank you for your comment. As you say, women without habitual miso consumption tended to have poorer diets, which might contribute to sarcopenia. Thus, we have mentioned this point as one of the limitations of this study in the Discussion section described as below.

Discussion (Line 279-281)

“Fourth, women without habitual miso consumption might have poor quality of diet, thus, we cannot deny the possibility that the diet quality contributed to this result.”

Minor:

Point 4

Lines 156-163: Please include the n number along with the p-values. The percentages are misleading.

Response

Thank you for your suggestion. According to your suggestion, we have included the n number along with the p-values in the Results section described as below.

Results (Line 172-175)

“In addition, in women with habitual miso consumption, the proportions of patients with low skeletal muscle mass (35/133 [26.3 %] vs. 13/26 [50.0 %], p = 0.024), low muscle strength (60/133 [45.1 %] vs. 19/26 [73.1 %], p = 0.009), and sarcopenia (25/133 [18.8 %] vs. 11/26 [42.3 %], p = 0.018) were lower than those in women without habitual miso consumption.”

Reviewer 2 Report

I would like to thank the authors for an interesting article. Overall, I believe the article is in the scope of the journal and is an important piece of work. Please note my minor revisions below.

Line 37:

"The number of older patients with type 2 diabetes is increasing." - Statistic?

Line 78-79

"All the patients were questioned about the We defined" - please revise.

Line 140 

Please revise all units so they are formated e.g. kg/m2

Line 200

Discussion made regarding the high polyunsaturated fatty acids in soybean. Consider detailing further studies discussing relationship between omega-6/omega-3 and sarcopenia.

I would also consider the addition of this references to support findings herein:

Suthuvoravut U et al. (2020) Association between Traditional Japanese Diet Washoku and Sarcopenia in Community-Dwelling Older Adults: Findings from the Kashiwa Study Journal of Nutrituional Health Aging 24 3 282-289

Author Response

Point 1

Line 37: "The number of older patients with type 2 diabetes is increasing." - Statistic?

Response

Thank you for your suggestion. According to your suggestion, we have added the reference.

Introduction (Line 36-37)

“The number of older patients with type 2 diabetes (T2DM) in Japan is increasing, since the aging of the adult population increasing [1].”

Reference

  1. Charvat, H.; Goto, A;, Goto, M.; Inoue, M.; Heianza, Y.; Arase, Y.; Sone, H.; Nakagami, T.; Song, X.; Qiao, Q.; Tuomilehto, J.; Tsugane, S.; Noda, M.; Inoue, M. Impact of population aging on trends in diabetes prevalence: A meta-regression analysis of 160,000 Japanese adults. J. Diabetes Investig. 2015, 6, 533–542. doi: 10.1111/jdi.12333.

Point 2

Line 78-79: "All the patients were questioned about the We defined" - please revise.

Point 3

Line 140: Please revise all units so they are formated e.g. kg/m2

Response

Thank you for your comments. According to your comments we have revised them.

Point 4

Line 200: Discussion made regarding the high polyunsaturated fatty acids in soybean. Consider detailing further studies discussing relationship between omega-6/omega-3 and sarcopenia.

Response

Thank you for your variable suggestion. As you say, both omega-6 and omega-3 are negatively associated with sarcopenia. Thus, we have added this relationship in the Discussion section described as below.

Discussion (Line 246-250)

“The main ingredient of miso is soybeans. Soybeans contain high amounts of omega-3 and omega-6 fatty acids [50]. There was negative correlation between omega-3 fatty acids intake and the presence of sarcopenia in elderly patients with T2DM [51]. In addition, both omega-3 and omega-6 intakes are known to be important for muscle maintenance [52].”

Reference

  1. Dhakal, K. H.; Jung, K. H.; Chae, J. H.; Shannon, J. G.; Lee, J. D. “Variation of unsaturated fatty acids in soybean sprout of high oleic acid accessions.” Food chemistry vol. 164 (2014): 70-3.
  2. Okamura, T.; Hashimoto, Y.; Miki, A.; Kaji, A.; Sakai, R.; Iwai, K.; Osaka, T.; Ushigome, E.; Hamaguchi, M.; Yamazaki, M.; Fukui, M. “Reduced dietary omega-3 fatty acids intake is associated with sarcopenia in elderly patients with type 2 diabetes: a cross-sectional study of KAMOGAWA-DM cohort study.” J Clin Biochem Nutr. 2020, 66, 3, 233-237.
  3. Oh, S. L.; Lee, S. R.; Kim, J. S. “Effects of conjugated linoleic acid/n-3 and resistance training on muscle quality and expression of atrophy-related ubiquitin ligases in middle-aged mice with high-fat diet-induced obesity.” J Exerc Nutrition Biochem. 2017, 30, 21(3), 11-18.

Point 5

I would also consider the addition of this references to support findings herein:

Suthuvoravut U et al. (2020) Association between Traditional Japanese Diet Washoku and Sarcopenia in Community-Dwelling Older Adults: Findings from the Kashiwa Study Journal of Nutrituional Health Aging 24 3 282-289

Response

Thank you for your valuable suggestion. According to your suggestion, we have added this reference in the Discussion section described as below.

Discussion (Line 260-265)

“Thus, women without habitual miso consumption may have had poor diet qualities, which may have contributed to obesity and sarcopenia. Adherence to a dietary pattern rich in foods characteristic of the Japanese diet was associated with a lower prevalence of sarcopenia [58]. Miso soup is well known as a traditional Japanese food. Therefore, the consumption of miso soup contributes to the traditional Japanese diet, suggesting that women with habitual miso consumption have a lower prevalence of sarcopenia.”

Reference

  1. Suthuvoravut, U.; Takahashi, K.; Murayama, H.; Tanaka, T.; Akishita, M.; Iijima, K. “Association between Traditional Japanese Diet Washoku and Sarcopenia in Community-Dwelling Older Adults: Findings from the Kashiwa Study.” J Nutr Health Aging. 2020, 24(3), 282-289.

Reviewer 3 Report

In the paper entitled Habitual miso (fermented soybean paste) consumption is associated with a low prevalence of sarcopenia in patients with type 2 diabetes: A cross-sectional study Takashi and co workers investigated on miso consumption beneficial effect in terms of saropenia onset on a Japanese cohort of T2DM diagnosed patients: Authors highlighted that habitual miso eating is negatively related with sarcopenia presence in women but not in men.

Paper is well written, rationale is clear and tables are exhaustive. Authors’ conclusions are supported from data and sound logical.

Nevertheless some concerns are to be assessed to the authors.

  • In Introduction section row 58, Authors reported miso consumption is related to a lower risk of total mortality in Japanese population. This point deserves a better explanation.
  • In Material and Methods section, row 72 Authors should better explain exclusion criteria they used for this study. Since T2D is a very complex disease, authors should at least provide some information concerning patients’ therapy. Furthermore, in the same section row 78, Authors divided patients in smokers and non-smokers; since cigarette smoking has been closely associated with increased oxidative stress, could the authors explain the reason why smokers patients have been enrolled for this study?
  • In Result section row 178-180 Authors reported correlation results for miso consumption in women and man, however authors should add a new figure where those data are represented by using a correlation scatter plot in order to better explain these results.

Author Response

Point 1

In Introduction section row 58, Authors reported miso consumption is related to a lower risk of total mortality in Japanese population. This point deserves a better explanation.

Response

Thank you for your suggestion. Previous studies on the association between soy consumption and cardiovascular disease implied important roles for nutrients such as isoflavone, fiber, and potassium. Isoflavone was found to have blood pressure reducing and lipid profile improving properties in meta-analyses, whereas soy fiber was shown to lower cholesterol levels and induce weight loss in humans. Moreover, a prospective study in the United States showed that an increase in dietary fiber from beans was associated with reduced cardiovascular disease related death among women. Fermented soy products are richer in fiber and potassium as well as bioactive components than their non-fermented counterparts, including tofu. The bioactive components in fermented soy products include the fibrinolytic enzyme, nattokinase (in natto), and polyamine. The polyamine spermidine was previously found to have cardioprotective effects, with intake associated with reduced morality due to heart failure. Therefore, bioactive components and nutrients contained in fermented soy products might have multifaceted benefits for survival. According to your suggestion, we have added the explanation of the relationship between miso consumption and mortality risk in the Introduction section described as below.

Introduction (Line 55-63)

“Moreover, a recent study from Japan revealed that the intake of fermented soy products including miso, rather than soy products, was related to a lower risk of total mortality [23]. Furthermore, miso intake suppressed visceral fat accumulation and hepatic lipid deposits in mice [24]. Soybeans, which are the main ingredient of miso, contain bioactive substances such as isoflavones and dietary fiber, and are useful in the prevention of various diseases [25-27]. In addition, fermented soybean products contain more bioactive ingredients and nutrients than non-fermented soybean products [28,29]. Therefore, nutrients and bioactive components in fermented soybean products may have multifaceted benefits for health, including survival.”

Reference

  1. Nagata, C.; Wada, K.; Tamura, T.; Konishi, K.; Goto, Y.; Koda, S.; Kawachi, T.; Tsuji, M.; Nakamura, K. Dietary soy and natto intake and cardiovascular disease mortality in Japanese adults: the Takayama study. Am J Clin Nutr 2017, 105, 426-31. doi:10.3945/ajcn.116.137281
  2. Taku, K.; Lin, N.; Cai, D.; Hu, J.; Zhao, X.; Zhang, Y.; Wang, P.; Melby, M. K.; Hooper, L.; Kurzer, M. S.; Mizuno, S.; Ishimi, Y.; Watanabe, S. Effects of soy isoflavone extract supplements on blood pressure in adult humans: systematic review and meta-analysis of randomized placebo-controlled trials. J Hypertens 2010, 28, 1971-82. doi:10.1097/ HJH.0b013e32833c6edb
  3. Taku, K.; Umegaki, K.; Sato, Y.; Taki, Y.; Endoh, K.; Watanabe, S. Soy isoflavones lower serum total and LDL cholesterol in humans: a meta-analysis of 11 randomized controlled trials. Am J Clin Nutr 2007, 85,1148-56. doi:10.1093/ajcn/85.4.1148
  4. Okamoto, A.; Sugi, E.; Koizumi, Y.; Yanagida, F.; Udaka, S. Polyamine content of ordinary foodstuffs and various fermented foods. Biosci Biotechnol Biochem 1997, 61, 1582-4. doi:10.1271/bbb.61.1582
  5. Sumi, H.; Hamada, H.; Tsushima, H.; Mihara, H.; Muraki, H. A novel fibrinolytic enzyme (nattokinase) in the vegetable cheese Natto; a typical and popular soybean food in the Japanese diet. Experientia 1987, 43, 1110-1. doi:10.1007/BF0195605

Point 2

In Material and Methods section, row 72 Authors should better explain exclusion criteria they used for this study.

Response

Thank you for your comment. According to your comment, we have explained exclusion criteria in detail in the Materials and Methods section describes as below.

Materials and Methods (Line 72-73)

“Exclusion criteria were follows; no data of multifrequency impedance body composition analyzer, no data of hand grip strength, incomplete questionnaires, and patients without T2DM.”

Point 3

Since T2D is a very complex disease, authors should at least provide some information concerning patients’ therapy.

Response

Thank you for your comment. According to your comment, we have added the data to the table. Thus, we described in the Materials and methods section and modified the Table 1 and Table 3 as below.

Materials and Methods (Line 88-90)

“Additionally, the data on medications, including those for diabetes and hypertension (renin-angiotensin-aldosterone inhibitor) were obtained from the patients' medical records.”

Table 1. Clinical characteristics of study participants

All

n = 351

Men

n = 192

Women

n = 159

p

Insulin (-/+)

263/88

145/47

118/41

0.875

DPP4 inhibitor (-/+)

169/182

86/106

83/76

0.202

SU (-/+)

270/81

144/48

126/33

0.417

Glinide (-/+)

320/31

180/12

140/19

0.092

Biguanide (-/+)

207/144

118/74

89/70

0.352

Thiazolidinediones (-/+)

338/13

183/9

155/4

0.430

α- GI (-/+)

306/45

169/23

137/22

0.721

SGLT2 inhibitor (-/+)

294/57

160/32

134/25

0.926

GLP-1 receptor agonist (-/+)

296/55

167/25

129/30

0.176

Data was expressed as number. The difference between group was evaluated by chi-square test. α- GI, alpha-glucosidase inhibitor.

Table 3. Clinical characteristics according to habitual miso consumption

Men, n = 192

Women, n = 159

Habitual miso consumption (-)

n = 23

Habitual miso consumption (+)

n = 169

p

Habitual miso consumption (-)

n = 26

Habitual miso consumption (+)

n = 133

p

Insulin (-/+)

20/3

125/44

0.271

15/11

103/30

0.063

DPP4 (-/+)

9/14

77/92

0.720

12/14

71/62

0.645

SU (-/+)

13/10

131/38

0.054

19/7

107/26

0.559

Glinide (-/+)

21/2

159/10

0.954

23/3

117/16

1.000

Biguanide (-/+)

9/14

109/60

0.034

16/10

73/60

0.683

Thiazolidinediones (-/+)

22/1

161/8

1.000

25/1

130/3

1.000

α- GI (-/+)

18/5

151/18

0.232

24/2

113/20

0.496

SGLT2 (-/+)

16/7

144/25

0.112

23/3

111/22

0.729

GLP-1 (-/+)

18/5

149/20

0.320

18/8

111/22

0.155

Data was expressed as number. The difference between group was evaluated by chi-square test. α- GI, alpha-glucosidase inhibitor.

Point 4

Furthermore, in the same section row 78, Authors divided patients in smokers and non-smokers; since cigarette smoking has been closely associated with increased oxidative stress, could the authors explain the reason why smokers patients have been enrolled for this study?

Response

Thank you for your comment. As you say, cigarette smoking has been closely associated with increased oxidative stress and thus, there is a possibility that smoking has effect on the sarcopenia. However, previous studies included both patients with and without habit of smoking. Therefore, we also included the patients with habit of smoking in this study.

According to your comment, we have conducted an analysis, divided by smoking status. In men, the association between habitual miso consumption and the presence of sarcopenia was absent in men with (adjusted OR, 0.68 [95% CI: 0.02–29.90], p = 0.839), and without habit of smoking (adjusted OR, 1.26 [95% CI: 0.27–5.98], p = 0.768). In women without habit of smoking, habitual miso consumption was negatively related to the presence of sarcopenia (adjusted OR, 0.36 [95% CI: 0.14–0.94], p = 0.036), whereas habitual miso consumption was not associated with the presence of sarcopenia in women with habit of smoking (adjusted OR, 0.07 [95% CI: 0.00–3.70], p = 0.187). This might be due to the high oxidative stress caused by smoking. Thus, we have added these points in the Materials and methods, Results and Discussion sections described as below.

Materials and Methods (Line 133-134)

“We performed sub-analyses to examine the effects of habitual miso soup consumption on the prevalence of sarcopenia adjusted by age, according to the presence or absence of habit of smoking.”

Results (Line 219-225)

“In men, the association between habitual miso consumption and the presence of sarcopenia was absent in men both with (adjusted OR, 0.68 [95% CI: 0.02–29.90], p = 0.839), and without habit of smoking (adjusted OR, 1.26 [95% CI: 0.27–5.98], p = 0.768). On the other hand, in women without habit of smoking, habitual miso consumption was negatively related to the presence of sarcopenia (adjusted OR, 0.36 [95% CI: 0.14–0.94], p = 0.036), whereas habitual miso consumption was not associated with the presence of sarcopenia in women with habit of smoking (adjusted OR, 0.07 [95% CI: 0.00–3.70], p = 0.187).”

Discussion (Line 271-275)

“In women without habit of smoking, habitual miso consumption was associated with the presence of sarcopenia women with habit of smoking, but in women with habit of smoking, habitual miso consumption was not associated with the presence of sarcopenia. This is due to the high oxidative stress caused by smoking [59]. In fact, habit of smoking is known to be the risk factor of sarcopenia [60].”

Reference

  1. Zuo, L.; He, F.; Sergakis, G. G.; Koozehchian, M. S.; Stimpfl, J. N.; Rong, Y.; Diaz, P. T.; Best, T. M. Interrelated role of cigarette smoking, oxidative stress, and immune response in COPD and corresponding treatments. Am J Physiol Lung Cell Mol Physiol. 2014, 1, 307(3), L205-18.
  2. Rom, O.; Kaisari, S.; Aizenbud, D.; Reznick, A. Z. Sarcopenia and smoking: a possible cellular model of cigarette smoke effects on muscle protein breakdown. Ann N Y Acad Sci. 2012, 1259, 47-53.

Point 5

In Result section row 178-180 Authors reported correlation results for miso consumption in women and man, however authors should add a new figure where those data are represented by using a correlation scatter plot in order to better explain these results.

Response

Thank you for your valuable suggestion. According to your suggestion, we have added new figures about the miso soup intake and skeletal muscle mass index or adjusted grip strength in men and women, separately. In women, the miso soup intake was correlated with skeletal muscle mass index (r = 0.25, p = 0.003) and adjusted grip strength (r = 0.17, p = 0.030). However, the miso soup intake was not correlated with skeletal muscle mass index (r = 0.14, p = 0.079) and adjusted grip strength (r = 0.14, p = 0.061) in men. Thus, we have added the correlation of the miso soup intake and skeletal muscle mass index or adjusted grip strength in the Materials and Methods section and Result section described as below.

Materials and Methods (Line 122-123)

“The correlation was analyzed by Pearson’s correlation coefficient.”

Results (Line196-200)

“Figure 2 shows the correlation between the log (miso soup intake + 1) and skeletal muscle mass or muscle strength. In women, the log (miso soup intake + 1) was correlated with skeletal muscle mass index (r = 0.25, p = 0.003) and adjusted grip strength (r = 0.17, p = 0.030). However, the log (miso soup intake + 1) was not correlated with skeletal muscle mass index (r = 0.14, p = 0.079) and adjusted grip strength (r = 0.14, p = 0.061) in men.”